# Enhancing ERα-targeted compound efficacy in breast cancer threapy with ExplainableAI and GeneticAlgorithm

**Zeonlung Pun**[1]*, **Qiaoyun Xue**[2], **Yichi Zhang**[3]

**1** Department of Mathematics and Statistics, Huazhong Agricultural University, Wuhan 430000, China,
**2** Department of Mathematics and Statistics, University of Glasgow, Glasgow G128QQ, United Kingdom,
**3** Faculty of Economics, Liaoning University, Shenyang, 110000, China

\* zeonlungpun@gmail.com

**Data availability statement:** The data and code that support the findings of this study are availability at github: https://github.com/ZeonlungPun/compoundEvaluate

## Abstract

Breast cancer remains a major cause of mortality among women globally, driving the need for advanced therapeutic solutions. This study presents a novel, comprehensive methodology integrating explainable artificial intelligence (AI), machine learning models, and genetic algorithms to enhance the bioactivity and ADMET (Absorption, Distribution, Metabolism, Excretion, and Toxicity) properties of compounds targeting estrogen receptor alpha (ERα). By employing SHAP (SHapley Additive exPlanations) and LassoNet, we identified and refined 50 critical molecular descriptors from an initial set of 729, significantly influencing the prediction of bioactivity. The selected descriptors were systematically validated, bolstering the predictive robustness of our models, which demonstrated a mean coefficient of determination of 77% for bioactivity and high accuracy scores of 90.2%, 93.7%, 89.5%, 87.3%, and 95.8% for absorption, distribution, metabolism, excretion, and toxicity, respectively. Further optimization through genetic algorithms identified candidate compounds with superior bioactivity, achieving $pIC_{50}$ values as high as 10.05, surpassing the previously observed peak values in the dataset. These results underscore the potential of leveraging advanced machine learning and optimization techniques to accelerate the discovery of effective cancer therapies.

## Introduction

Breast cancer remains one of the most prevalent and deadly cancers worldwide. A critical factor in the progression of breast cancer is the estrogen receptor alpha (ERα), which plays a pivotal role in tumor development. ERα is predominantly expressed in a small subset of luminal cells, representing less than 10% of normal mammary epithelial cells. However, its expression increases dramatically in breast tumors, with over 70% of breast cancers showing high levels of ERα (ER+ tumors). Despite its significant presence in cancerous tissues, the mechanisms underlying the selective expression of ERα in normal versus cancerous mammary cells remain poorly understood [1]. Given its critical role in breast cancer, ERα has become a key target for therapeutic interventions. Compounds capable of inhibiting ERα activity have shown considerable promise as potential treatments for ER+ breast cancer.

**Funding:** The author(s) received no specific funding for this work.

**Competing interests:** The authors have declared that no competing interests exist.

The development of pharmaceutical compounds relies heavily on accurately predicting their bioactivity. Compounds that antagonize ERα activity are potential candidates for the treatment of breast cancer. In this study, we use the $IC_{50}$ value as a measure of bioactivity. This experimentally determined value (measured in nM) reflects the concentration required to inhibit ERα activity by 50%. A smaller $IC_{50}$ value indicates higher biological activity and greater effectiveness in inhibiting ERα activity, making it a critical parameter for evaluating potential drug candidates.

In addition to bioactivity, predicting ADMET (Absorption, Distribution, Metabolism, Excretion, and Toxicity) properties is equally essential. These pharmacokinetic and safety attributes are fundamental to assessing a compound's suitability for drug development. To streamline this process and reduce costs, researchers frequently employ quantitative structure-activity relationship (QSAR) models. QSAR models enable the prediction of bioactivity and ADMET properties based on molecular descriptors, thus playing a pivotal role in modern drug discovery.

QSAR represents an evolution of the traditional Structure-Activity Relationship (SAR) analysis. The roots of SAR can be traced back to the late 19th century when chemists first began exploring the relationship between the structure of chemical compounds and their biological activity. However, SAR methods primarily relied on the experience and intuitive judgment of chemists and lacked a formalized and quantitative framework. In contrast, QSAR models employ statistical methods to quantitatively analyze the relationship between molecular structure and biological activity. Early QSAR models were predominantly linear, with one of the most well-known being the Hansch Model [2]. Hansch demonstrated that the biological activity of a molecule is largely influenced by its hydrophobic effect ($\log P$), steric effect ($E_s$), and electrostatic effect ($\sigma$), under the assumption that these effects can be added independently. Based on this premise, the Hansch model was formulated as a linear regression equation:

$$\log\left(\frac{1}{C}\right) = a \log P + bE_s + c\sigma + d$$

where $a$, $b$, $c$, and $d$ are the coefficients representing the contribution of each individual effect to the overall biological activity.

QSAR models are vital in drug development, exploiting the principle that structural variations in molecules influence biological effects [3]. This approach is particularly useful for discovering new molecules and refining existing ones. Despite their utility, traditional QSAR methods have certain limitations, especially in complex fields such as cancer research. One major limitation is that many QSAR models assume a linear relationship between molecular descriptors and biological activity. However, this assumption overlooks the more nuanced molecular interactions that are crucial in complex biological systems, such as those involved in cancer biology. Furthermore, traditional QSAR models often lack interpretability, which is essential for understanding how molecular features impact biological activity. This lack of transparency can hinder the optimization of potential treatments. Additionally, while QSAR models can address the relationship between chemical structure and bioactivity, they often do not comprehensively cover ADMET properties. This gap can result in high attrition rates in drug development, as compounds may fail to meet necessary pharmacokinetic and safety standards. Another challenge with traditional QSAR methods is their susceptibility to overfitting, especially in cases where the descriptor data is highly dimensional. This issue arises when too many descriptors are used without proper selection and validation, leading to poor predictive performance.

For a compound to advance as a viable drug candidate, it must also exhibit favorable ADMET properties. These properties dictate how a compound is absorbed, distributed, metabolized, and excreted by the body, and they define its potential toxicity. Even if a compound shows high biological activity, inadequate ADMET characteristics—such as poor absorption, rapid metabolism, or excessive toxicity—can impede its development into a practical drug. Therefore, optimizing ADMET properties is a critical step in the drug development pipeline, ensuring that promising bioactive compounds can transition into safe and effective drugs [4].

In this paper, we propose a streamlined workflow to predict the bioactivity and ADMET properties of compounds. Our comprehensive pipeline integrates the strengths of Explainable AI and machine learning models for virtual drug screening. Rather than using these methods in isolation, we combine their predictive results with a Genetic Algorithm to evaluate each candidate compound, providing valuable insights for drug design research. Our objective is to thoroughly assess the potential of these compounds as candidates for anti-breast cancer drugs. Initially, we employ explainable AI to filter chemical descriptors, which are essential for building accurate predictive models and reducing computational costs. The selected descriptors are then evaluated for their suitability and used to construct both bioactivity regression models and ADMET classification models. Simultaneously, we apply Genetic Algorithms in conjunction with our predictive models to identify chemical descriptor values that maximize bioactivity. This integrated approach allows us to identify optimal candidate compounds from our dataset, enhancing the prospects of discovering effective anti-breast cancer drugs.

The structure of this paper is organized as follows: Section 2 discusses related works in QSAR modeling and ADMET predictions, providing context and illustrating how our approach integrates and expands upon existing research. Section 3 outlines the datasets used and details our proposed workflow, highlighting the innovative aspects of our methodology. Section 4 presents the results of our predictive models, analyzing their effectiveness in identifying promising drug candidates. Section 5 concludes with a discussion of the broader conclusions, implications of our findings, and limitations of our study, while also suggesting directions for future research.

## Related works

Recent advancements in computer technology have significantly enhanced the adoption of data-driven models especially Machine Learning models for predicting compound bioactivity. Hamzehali et al. [5] utilized QSAR models incorporating the Monte Carlo algorithm and the balance correlation method via CORAL software to estimate the inhibition potencies of various imatinib derivatives targeting BCR-ABL TK. While effective, this approach relies heavily on specific software, and the Monte Carlo estimation introduces considerable uncertainty.

In contrast, Daoui et al. [6] explored 48 novel 4,5,6,7-tetrahydrobenzo[D]-thiazol-2 derivatives using both linear and nonlinear regression techniques, as well as artificial neural networks, to inhibit the c-Met receptor tyrosine kinase. Further diversifying modeling approaches, Kwon et al. [3] employed an ensemble method that integrates various models via meta-learning, alongside an end-to-end neural network that autonomously extracts features from SMILES data.

Adding to the discourse, Hsiao et al. [7] confirmed the effectiveness of popular machine learning techniques by analyzing a publicly available in vivo dataset, underscoring the robustness of these methods in ADMET modeling. Despite these advancements, the integration of bioactivity prediction with ADMET modeling remains a challenge. Additionally, the high

accuracy of these models often comes at the cost of decreased interpretability, attributed to their 'black-box' nature.

Li et al. [8] applied three machine learning algorithms to develop models for predicting the ADMET properties of anti-breast cancer compounds. Their results indicate that the Light-GBM model can reliably predict molecular ADMET properties. Although their work is note-worthy, it lacked a combined approach for ADMET classification and other prediction models. Furthermore, their model utilized all molecular descriptors to train the model, which may not be practical for real-world applications due to computational constraints.

Wu et al. [9] introduced a novel approach using LightGBM as a variable selection model to assess descriptor importance, selecting the top 20 most significant descriptors to build predictive models for biological activity and ADMET properties. However, they did not provide detailed criteria for the number of descriptor selection or validate the appropriateness of the chosen descriptors. Furthermore, their BVAP method, which simply combines predicted activity values with ADMET properties, lacks detailed application and explanation.

In summary, while most existing works have effectively leveraged machine learning models to predict the bioactivity and ADMET properties of various compounds, the application of the latest explainable technologies is relatively lacking. Although some studies, such as Wu et al. [9], have identified the top 20 molecular descriptors with higher importance, they do not provide clear criteria for distinguishing between vital and less important descriptors, nor do they specify the exact number of vital descriptors. Furthermore, there is a need for a recognized algorithm to assess and quantify the potential of each candidate compound.

To address existing research gaps, we propose a comprehensive modeling pipeline that integrates machine learning models with advanced explainable AI technologies. Our methodology is structured into four distinct phases, as shown in Fig. 1:

- **Descriptor analysis**: We begin by employing SHAP (SHapley Additive exPlanations) [10] and LassoNet [11] to analyze the importance of each descriptor. These insights guide us in selecting an optimal number of descriptors for further analysis. The efficacy of our selection is validated through a method we term 'independent variable perturbation analysis'.
- **Bioactivity prediction**: Using the validated descriptors, we construct a LightGBM regression model to predict the bioactivity of compounds. This model is then subjected to sensitivity testing.
- **ADMET property prediction**: Simultaneously, we employ the same set of descriptors to develop an XGBoost classification model focused on predicting the ADMET properties of the compounds.
- **Model integration and optimization**: In the final phase, we explore the synergy of combining the regression and classification models using Genetic Algorithms. This integration aims to identify descriptor values that maximize biological activity and pinpoint the most promising drug candidate compounds.

## Materials and methods

### Dataset

The dataset employed in this study consists of 1,974 compounds identified as potential therapeutic agents against breast cancer, specifically by antagonizing the activity of estrogen receptor alpha (ERα). This dataset was provided by the China Association for Science and Technology and originates from Question D of the 18th Chinese Graduate Mathematical Modeling Contest (Readers can access it on our GitHub repository in Data Availability Statement

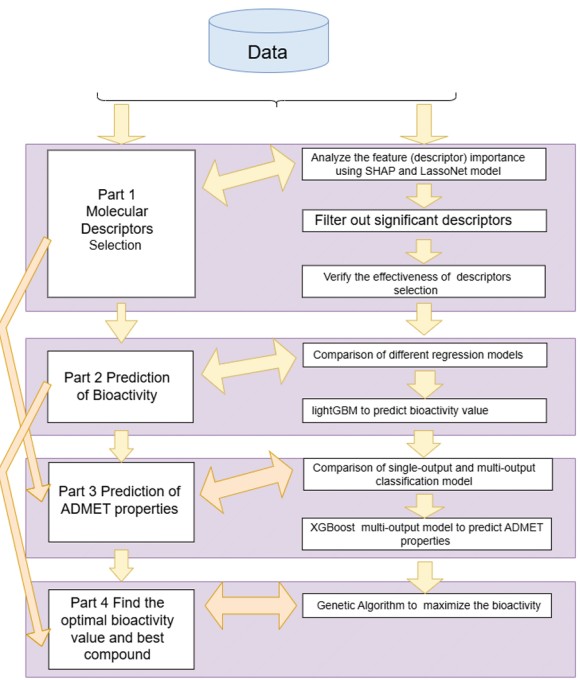

**Fig 1. The flowchart of this paper.**

or via the link provided in Data Availability Statement.). It includes both the bioactivity values and the ADMET properties of the compounds. Each compound is characterized by 729 molecular descriptors, encompassing a range of structural and physicochemical properties such as molecular weight, LogP (a measure of lipophilicity), and topological characteristics like the number of hydrogen bond donors and acceptors. The bioactivity against ERα is quantitatively assessed using the $pIC_{50}$ value, which represents the negative logarithm of the $IC_{50}$ value. The $pIC_{50}$ value is typically positively correlated with biological activity, meaning that a higher $pIC_{50}$ indicates greater inhibitory effectiveness. In practical QSAR modeling, $pIC_{50}$ is generally used to represent the biological activity value.

To facilitate modeling and prediction, this study focuses on five key ADMET properties of the compounds in the dataset, evaluated using a binary classification method. These properties are defined as follows:

- **Caco-2 permeability** ($Y_1$): This property measures the permeability of a compound across small intestinal epithelial cells. A value of '1' indicates good permeability, suggesting effective absorption in the human gastrointestinal tract, whereas a value of '0' denotes poor permeability.
- **CYP3A4 metabolism** ($Y_2$): This property assesses whether a compound is metabolizable by the CYP3A4 enzyme, a key metabolic enzyme in humans. A value of '1' indicates that the compound is metabolizable by CYP3A4, while '0' indicates it is not, suggesting potential issues with metabolic stability.
- **hERG channel interaction** ($Y_3$): This property evaluates the cardiotoxic potential of a compound through its interaction with the human Ether-a-go-go Related Gene (hERG) channel. A value of '1' denotes cardiotoxic potential, which could pose safety concerns, while a value of '0' suggests an absence of such toxicity.

- **Human oral bioavailability (HOB, $Y_4$)**: This property measures the proportion of the drug that enters systemic circulation when administered orally. A value of "1" represents good oral bioavailability, indicating effective absorption into the bloodstream, while a value of "0" denotes poor oral bioavailability.
- **Mutagenicity (MN, $Y_5$)**: This property is assessed through the micronucleus test, which evaluates whether a compound has genotoxic potential. A value of "1" indicates that the compound is genotoxic, posing potential risks of genetic damage, while a value of "0" indicates non-genotoxicity.

It is worth noting that according to the type of descriptor, we can roughly divide them into 53 groups, as shown in Table 7.

## Methods

In this section, we propose a comprehensive workflow that integrates machine learning models and Genetic Algorithms to assess the importance of each Molecular Descriptor, predict the bioactivity and ADMET properties of each component, and identify the most promising compounds for developing drugs for breast cancer therapy.

Compared to traditional QSAR linear models, machine learning models excel in their ability to capture complex, nonlinear relationships between Molecular Descriptors and the bioactivity of each component, thereby achieving higher predictive accuracy. Additionally, the use of explainable AI algorithms enables us to interpret the prediction process, providing insights that break the "black box" nature of machine learning models and enhancing their reliability in drug discovery. The Genetic Algorithm plays a crucial role in this workflow, as it allows the optimization process to consider both bioactivity and ADMET properties simultaneously. By combining the predictions of bioactivity and ADMET models, Genetic Algorithms ensures that the optimal potential compounds identified are not only highly bioactive but also exhibit favorable ADMET profiles, making them more suitable for drug development.

**Descriptor selection with explainable AI model.** Explainable AI technology fulfills two primary roles in this study. Firstly, it aids in eliminating redundancy from the dataset by identifying descriptors that are essential for predicting bioactivity, thereby effectively reducing feature dimensions. This screening not only streamlines the dataset but also enhances the rationality of subsequent modeling efforts. Secondly, it increases the transparency of machine learning models, making them easier to comprehend and evaluate. In our research, we employ two well-established methods: the SHAP model and the LassoNet model. In our study, SHAP and LassoNet play complementary roles in descriptor selection. SHAP identifies individual molecular descriptors with the highest impact on bioactivity predictions, while LassoNet evaluates descriptor groups, considering their interactions and collective contribution to model performance. Together, these two methods can complement one another.

Additionally, while classical feature dimension reduction methods such as Principal Components Analysis (PCA) and t-Distributed Stochastic Neighbor Embedding (t-SNE) are commonly used, they do not maintain the original features, rendering them less convenient for subsequent analysis and utilization.

SHAP is a widely-used method derived from cooperative game theory, designed to explain the output of machine learning models. It provides insights into how individual features contribute to a model's predictions by calculating the marginal contribution of each feature across all possible feature combinations. In this study, SHAP values are utilized to assess the importance of molecular descriptors by quantifying their impact on bioactivity predictions. A higher SHAP value for a feature indicates a greater influence on the model's predictions,

enabling us to identify the most critical descriptors for building accurate models. These values quantify the impact of each feature on the model's predictions by evaluating the change in prediction outcome when a feature is added to different combinations of other features [10]. This assessment is visualized through a feature importance plot, where features are ranked based on their SHAP values. Features with greater impacts on the model's predictive accuracy are assigned higher importance rankings, facilitating a clearer understanding of model behavior.

Specifically, for a set of all features $F$ and a subset $S \subseteq F$, the Shapley value of feature $i$ is computed as follows:

$$\phi_i = \sum_{S \subseteq F \setminus i} \frac{|S|!(|F| - |S| - 1)!}{|F|!} [f_{S \cup i}(X_{S \cup i}) - f_S(X_S)] \tag{1}$$

In our implementation, we utilize the TreeExplainer module from the SHAP package [12]. For each descriptor, the TreeExplainer assigns a SHAP value to every sample in the training set. We then take the absolute value of each SHAP value and compute the average of these absolute values to serve as the final evaluation score for each descriptor.

The SHAP model, while effective in assessing the importance of individual descriptors, evaluates each descriptor in isolation, potentially overlooking their interactions. As indicated in Table 7, descriptors can be categorized into groups, suggesting that evaluating the importance of descriptor groups simultaneously could yield a more comprehensive understanding. To accommodate this, the group lasso model [13] could be employed as a supplementary method. This model is adept at managing grouped features but typically assumes a linear relationship between independent and dependent variables, which may not be suitable in complex scenarios with nonlinear relationships.

To better handle nonlinear relationships, we incorporate LassoNet, a hybrid model that combines the classic LASSO approach [14] with neural network architecture. LassoNet is an advanced feature selection method that integrates the strengths of neural networks and the classic LASSO regression. It combines feature sparsity (achieved through the LASSO approach) with the flexibility of neural network architectures. A unique aspect of LassoNet is its "skip layer mechanism," which ensures that features are retained in deeper layers of the network only if they are deemed relevant at the initial stage. This makes LassoNet particularly effective in handling complex, non-linear relationships between descriptors and bioactivity. In this study, LassoNet is employed to identify groups of descriptors that collectively contribute to predicting bioactivity, offering a complementary perspective to SHAP's individual feature assessment.The model is formalized as:

$$\min_{\theta, W} L(\theta, W) + \lambda \|\theta\|_1 \tag{2}$$

$$\text{s. t.} \quad \|W_j^{(1)}\|_\infty \le M|\theta_j|, \text{ for } j = 1, \cdots, d. \tag{3}$$

where $L(\theta, W)$ represents the loss function, and $W_j^{(1)}$ denotes the weights for feature $j$ in the first hidden layer. In our implementation, we construct a neural network with 2 hidden layers and set $M$ to 10. Other hyper parameters are set as defaults.

This study employs SHAP and LassoNet to analyze and affirm the significance of various descriptors in predicting the bioactivity of compounds. We will now discuss the detailed steps involved in evaluating each descriptor and selecting the most critical ones for effective modeling.

Initially, the process begins by determining the optimal number of descriptors to retain, a crucial step for effective dimension reduction. We organize a candidate descriptor set $C$ by sorting descriptors from largest to smallest based on their importance scores as calculated by the SHAP model. To ascertain the most effective subset, we sequentially construct regression models with varying numbers of top descriptors: starting with 2, then increasing to 5, 8, and continuing up to 290. The performance of each model iteration is evaluated using the Coefficient of Determination ($R^2$) and the mean squared error (MSE). To ensure the reliability and robustness of our findings, we replicate the modeling process 20 times for each descriptor set.

Upon determining the optimal number of descriptors, the next phase involves selecting the most influential descriptors by integrating results from both the LassoNet and SHAP models. We specifically select the two descriptors with the highest scores from each of the five most important groups identified by LassoNet. The remaining descriptors are chosen based on their high scores from the SHAP model, ensuring that our selection includes the most predictive features for modeling the bioactivity of compounds.

**Verify the selected descriptors.** To verify the effectiveness and rationality of our selected descriptors, we designed a set of comparative experiments. These experiments involve permuting the variables in each control group according to their importance scores and then modeling. We term this method "independent variable perturbation analysis."

The experiments are structured as follows:

- **Control Group 1**: Models using the top 50 deteriorates with the highest importance scores.
- **Control Group 2**: Models using 50 deteriorates with importance scores ranked from 51 to 100.
- **Control Group 3**: Models using 50 deteriorates with importance scores ranked from 101 to 150.
- **Control Group 4**: Models using 50 randomly selected deteriorates from the entire set.
- **Control Group 5**: Models using 50 deteriorates randomly selected from the complement set of $C$—those deemed less important by both the SHAP model and LassoNet.

Each modeling scenario is replicated 20 times to ensure statistical reliability, and the results are compared using the test set data.

**Prediction of bioactivity.** This study evaluate several machine learning models to predict the bioactivity of chemicals based on their descriptors, including Multi-Layer Perceptron (MLP), Support Vector Machine (SVM), LightGBM (LGB) [15], XGBoost [16], and Random Forest (RF) [17].

MLP, a neural network-based model, excels at capturing complex non-linear relationships, making it suitable for high-dimensional molecular descriptors, though it requires large amounts of data to avoid overfitting. SVM, on the other hand, is well-suited for small datasets and performs effectively in high-dimensional spaces using kernel-based methods, though its computational cost increases significantly with larger datasets. XGBoost, a tree-based ensemble method, is recognized for its efficiency and accuracy on tabular data. It provides robust performance with built-in feature importance ranking, but its reliance on well-engineered features can limit performance in less-structured data. Similarly, RF, another tree-based ensemble model, is highly resilient to overfitting and performs well on heterogeneous data by aggregating predictions from multiple decision trees, although it may struggle to model complex

interactions compared to gradient boosting methods like XGBoost. LGB, a gradient boosting framework developed by Microsoft, is renowned for its efficiency, scalability, and accuracy. By utilizing techniques such as Gradient-based One-Side Sampling and a histogram-based approach for tree construction, LGB optimizes memory usage and significantly reduces training time, making it highly effective for handling large datasets.

By leveraging these models, we aim to evaluate their comparative performance in predicting bioactivity and to identify the most effective approach for our dataset. Based on the comparison results of different models presented in Fig 2 in the subsequent section, LGB is selected for further analysis due to its superior performance and computational efficiency.

In addition, we will also compare our proposed workflow with five alternative methods from the literature. The details of each method are as follows:

- **Method 1** [9]: This method uses LightGBM to select the 20 most important molecular descriptors and then employs a neural network with two hidden layers to model the relationship between these descriptors and bioactivity.
- **Method 2** [18]: This approach directly utilizes all available molecular descriptors and applies a Gradient Boosting method to predict bioactivity.
- **Method 3 (Hansch Model/Linear Model)**: In this baseline approach, we extend the traditional Hansch Model [2] by utilizing all available descriptors to fit a linear regression model, rather than restricting the features to those originally used in the Hansch framework. This serves as a baseline for evaluating other methods.
- **Method 4**: This method involves two steps. First, LASSO (Least Absolute Shrinkage and Selection Operator) is used to select the 50 most significant descriptors. Then, a linear regression model is applied to predict bioactivity.
- **Method 5** [6]: This approach uses PCA to reduce the dimensionality of the input features. Specifically, they retain 15 Principal Components in the final model. Subsequently, a neural network with a single hidden layer is employed to model the relationship between these principal components and bioactivity.

By comparing these methods, we aim to demonstrate the advantages of our workflow in terms of predictive performance and robustness in handling molecular descriptors.

**Prediction of ADMET properties.** In this phase of our study, we focus on developing a classification model to accurately predict the ADMET properties of various compounds. We explore both single-output and multi-output classification strategies, employing LGB for the former and XGBoost for the latter. Constructing a single-output model with LGB requires the development of five distinct models, one for each ADMET property, which can be operationally cumbersome.

Upon comparative analysis of the predicted results from both approaches, we observed minimal performance differences between the single-output models and the multi-output model. Based on these findings, we chose to implement XGBoost for its efficiency in constructing a multi-output classification model. This approach not only streamlines the modeling process but also maintains high accuracy in predicting multiple ADMET properties simultaneously. As the study involves five binary classification tasks, the classification thresholds for all tasks were set at 0.5. Specifically, if the predicted probability exceeds 0.5, the instance is classified as class "1"; otherwise, it is classified as class "0."

**Finding optimal descriptor value and the best compound.** Genetic algorithms (GA) are robust search techniques derived from the principles of natural selection and genetics,

widely used to discover high-quality solutions to optimization problems [19]. These algorithms utilize biologically inspired operations such as mutation, crossover, and selection to evolve solutions towards optimal outcomes.

In this study, we employ GA to optimize the descriptors of compounds, with the goal of maximizing bioactivity while adhering to ADMET safety constraints. This optimization integrates two key components: the trained bioactivity prediction model $\hat{f}$, and the ADMET property prediction model $\hat{g}$. The optimization framework is formulated as follows:

$$\hat{x} = \arg max \, \hat{f}(x) \tag{4}$$

$$\text{s.t.} \quad \sum \hat{g}(x) \geq t \tag{5}$$

Here, $x$ represents the normalized descriptor values, ranging from [0,1]. The parameter $t$ sets the threshold for ADMET properties, where a higher value indicates more stringent safety requirements for the compound.

## Results

### Evaluation metrics

To evaluate the performance of the machine learning models used in this study, we employed two sets of metrics tailored to regression and classification tasks. For regression models, the metrics included the coefficient of determination ($R^2$) and the mean squared error (MSE). For binary classification models, we used accuracy (ACC) and the area under the receiver operating characteristic curve (ROC-AUC). These metrics provide complementary insights into the models' predictive accuracy and reliability.

$R^2$ **(Coefficient of determination):** The $R^2$ value measures the proportion of variance in the dependent variable that is explained by the independent variables in the model. It is calculated as:

$$R^2 = 1 - \frac{\sum_{i=1}^{n}(y_i - \hat{y}_i)^2}{\sum_{i=1}^{n}(y_i - \bar{y})^2}$$

where $y_i$ is the observed bioactivity value, $\hat{y}_i$ is the predicted bioactivity value, and $\bar{y}$ is the mean of the observed bioactivity values. An $R^2$ value closer to 1 indicates better model performance, as it implies that the model explains most of the variability in the data.

**MSE (mean squared error):** MSE quantifies the average squared difference between predicted and actual bioactivity values in regression tasks. It is computed as:

$$\text{MSE} = \frac{1}{n} \sum_{i=1}^{n}(y_i - \hat{y}_i)^2$$

where $n$ is the number of samples, $y_i$ is the observed value, and $\hat{y}_i$ is the predicted value. A lower MSE indicates better predictive accuracy, as it minimizes the magnitude of the prediction errors.

**Accuracy (ACC):** For binary classification tasks in ADMET properties prediction , Accuracy measures the proportion of correctly predicted instances among all samples. It is calculated as:

$$\text{Accuracy (ACC)} = \frac{\text{Number of Correct Predictions}}{\text{Total Number of Predictions}}$$

Accuracy provides a straightforward evaluation of the model's overall correctness. However, it can be misleading in imbalanced datasets, where one class may dominate the predictions.

**ROC-AUC (receiver operating characteristic - area under the curve)** ROC-AUC evaluates the model's ability to distinguish between positive and negative classes across various classification thresholds. The ROC curve plots the true positive rate (TPR) against the false positive rate (FPR), and the area under this curve (AUC) provides a single scalar value representing model performance. An AUC of 1.0 indicates perfect classification, while an AUC of 0.5 suggests no discriminatory power. ROC-AUC is particularly valuable in evaluating imbalanced datasets, as it considers both sensitivity and specificity, offering a more robust measure of classification performance compared to accuracy alone.

These metrics were selected to provide comprehensive evaluations of the models' performances, considering both regression and classification tasks. While $R^2$ and MSE focus on regression accuracy and error, ACC and ROC-AUC offer complementary insights into the effectiveness of binary classification models.

## Prediction of bioactivity

To develop a predictive model for compound bioactivity based on molecular descriptors, we employ a systematic approach comprising several key stages. Initially, we assess a variety of widely-used machine learning algorithms. After considering factors such as inference time and prediction accuracy, we select the LightGBM (LGB) model due to its superior performance. Subsequently, we utilize SHAP and LassoNet to determine the importance of each descriptor. This analysis is instrumental in identifying critical descriptors. Further, we analyze the curve of MSE and $R^2$ change with the number of important variables used'. This analysis is crucial as it helps in determining the optimal number of descriptors to use, ensuring that our model is both efficient and effective by confirming the significance of these selected descriptors. Finally, we perform a sensitivity analysis to examine the correlation between selected molecular descriptors and bioactivity.

**Evaluation of regression model.** We allocate the dataset into training, validation, and test subsets with respective ratios of 0.7, 0.15, and 0.15. The performance of five machine learning models—MLP, SVM, XGBoost, LGB, and RF—is evaluated. This evaluation involve conducting 20 repeated trials, training the models on the training set, and assessing their efficacy using the test set, as illustrated in Fig 2 and Table 1.

Analysis of the results demonstrated that MLP achieved the poorest performance, with an $R^2$ mean of only 63.7%. As discussed in the previous section, MLP requires a larger amount of data to perform effectively and often underperforms compared to tree-based ensemble models on tabular data [20]. Tree-based ensemble models, such as LGB and RF, are particularly well-suited for handling molecular descriptor data due to several key advantages. These models inherently manage high-dimensional molecular descriptor data without the need for explicit feature scaling or normalization. Additionally, they effectively capture non-linear interactions between molecular descriptors, which are critical for accurately predicting bioactivity. Furthermore, tree-based models are robust to irrelevant or redundant features, a common characteristic of molecular descriptor datasets, ensuring better generalization and reducing overfitting risks.

In contrast to MLP, the tree-based ensemble models, LGB and RF, produced the most favorable outcomes, with $R^2$ means of 77.6% and 76.1%, respectively. Among these, the LGB model was selected for the development of a predictive tool for bioactivity based on compound descriptors, owing to its superior performance, shorter inference time, and reduced

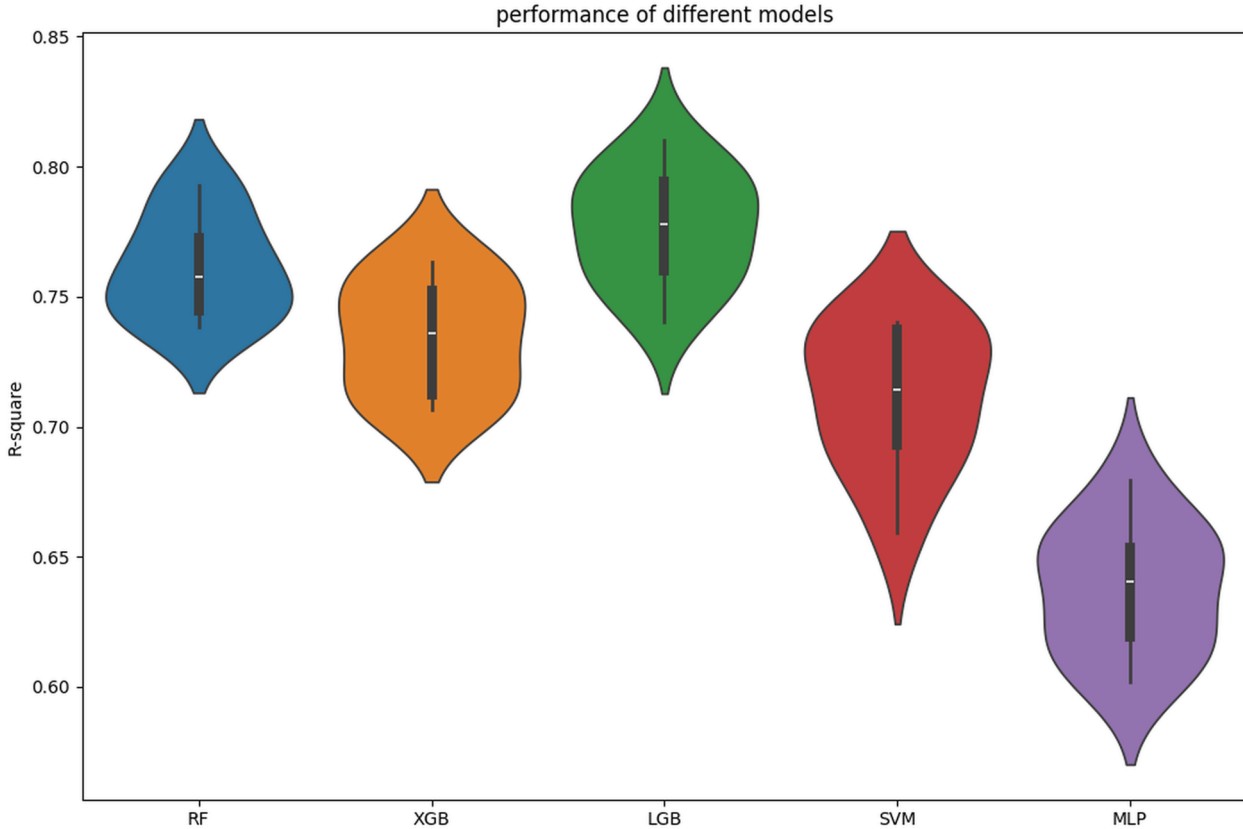

**Fig 2. The performance of different regression models.**

**Table 1. Summary statistics for different regression models.**

|  | RF | XGB | LGB | SVM | MLP |
|---|---|---|---|---|---|
| $R^2$ Mean | 76.1 % | 73.4% | 77.6% | 71.1% | 63.7% |
| $R^2$ Std | 0.020 | 0.022 | 0.021 | 0.027 | 0.025 |

training time compared to RF. This makes LGB particularly advantageous for large-scale bioactivity prediction tasks where computational efficiency is critical.

**Evaluate descriptor importance.** Using the LGB model combined with the SHAP variable selection model, we evaluate the importance of each descriptor. As shown in Fig 3, the SHAP model identifies descriptors such as MDEC-23, Lipoaffinity, maxHsOH, and nC as having a significant impact on predicting the bioactivity of compounds. Additionally, the SHAP model indicates that 291 descriptors are important for predicting bioactivity, each having non-zero weight scores. However, it is more reasonable to use descriptor types as sets of variables to evaluate their impact on bioactivity prediction as a supplement. Therefore, we also use LassoNet to evaluate the importance of each set of variables (descriptor types), as shown in Fig 4.

LassoNet identifies Atom Count, Chi Path, Extended Topochemical Atom, Molecular Distance Edge, and Ring Count as the five most important descriptor types. Atom Count quantifies the number of different types of a in a molecule, directly influencing its size, shape, and chemical properties. Chi Path captures the molecular topology and connectivity between

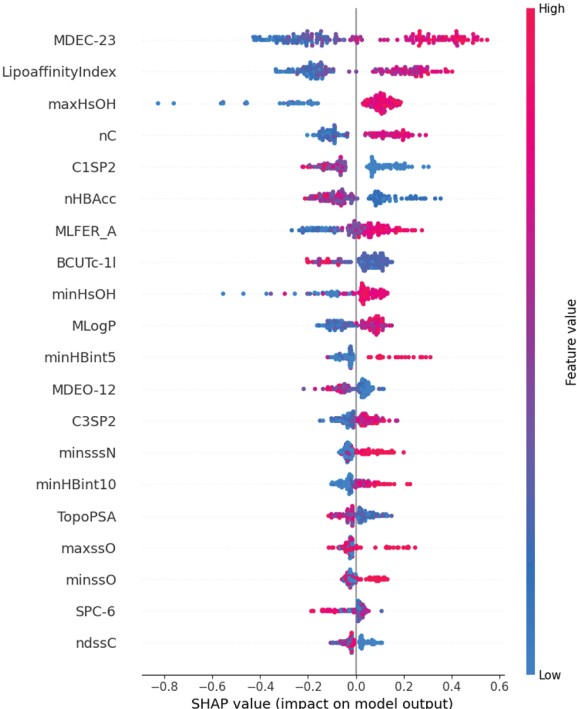

**Fig 3. Some visualized results of explainable AI models.** The 20 key descriptors identified by SHAP model.

atoms, revealing structural features critical for bioactivity. Extended Topochemical Atom descriptors combine atomic properties with topological information, providing a detailed view of the molecular environment and interactions. Molecular Distance Edge measures the distances between atoms within the molecule, affecting its three-dimensional conformation and interaction with ERα. These descriptors collectively provide a comprehensive understanding of the structural and chemical features that drive ERα inhibition, enhancing the accuracy of QSAR models in predicting compound bioactivity.

Among the five most important groups of descriptors selected by LassoNet, each group of descriptors are represented by the three descriptors with the highest scores. These are intersected with the 291 non-zero descriptors selected by the SHAP model to form a candidate descriptor set $C$ for further processing.

**Descriptor selection.** We now determine the optimal number of descriptors for our models. The aggregated results of this analysis are displayed in Figs 5 and 6 and Table 2. As expected, we observe a general trend where the MSE decreases and the $R^2$ increases with the addition of more descriptors. However, the inclusion of more than the top 50 descriptors does not significantly improve performance. Consequently, we have chosen to proceed with these 50 most significant descriptors for our subsequent modeling efforts.

Within this set of 50 descriptors, we specifically select the two descriptors with the highest scores from each of the five most important groups identified by LassoNet. The remaining 40 descriptors are selected based on their high scores from the SHAP model. The complete list of these selected descriptors is provided in Table 8.

**Verify the selected descriptors.** As illustrated in the boxplot in Fig 7, the performance of the control groups deteriorates sequentially from Group 1 to Group 5. This gradient in performance validates our model's evaluation method, confirming that the descriptors selected by

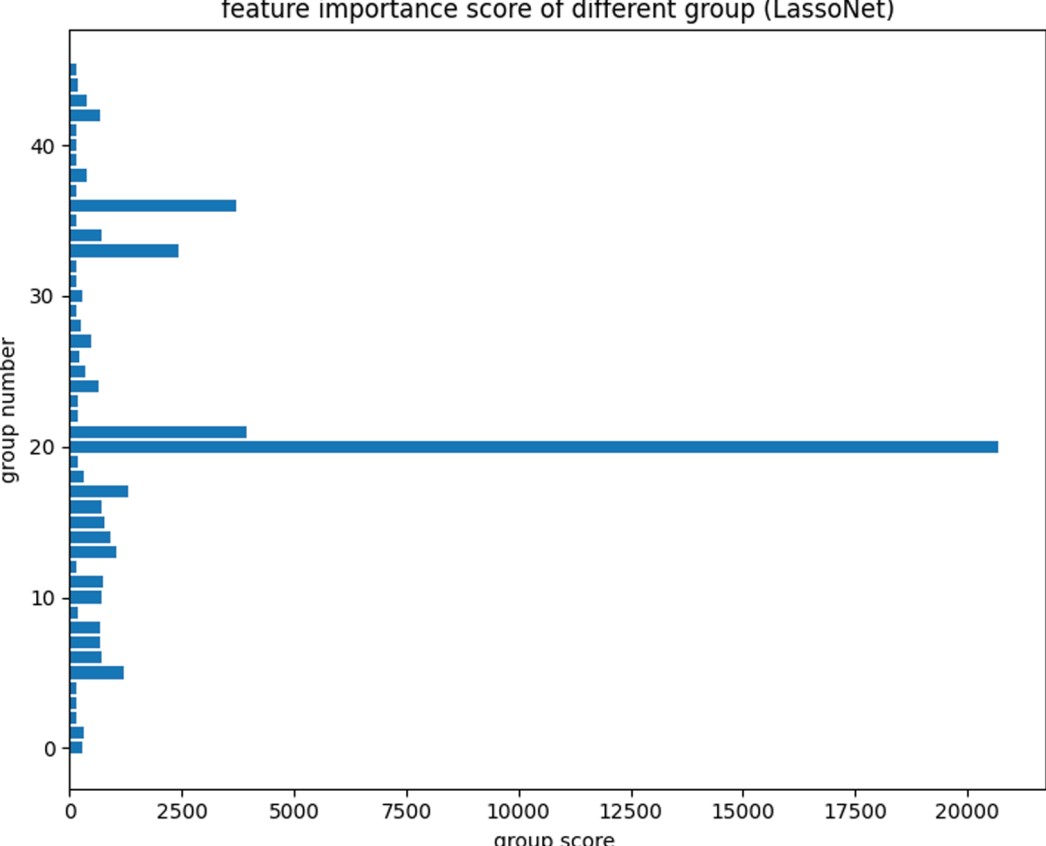

**Fig 4. Some visualized results of explainable AI models.** The importance score of each group calculated by the LassoNet model.

the model are indeed effective and reasonable. This outcome not only supports the robustness of our variable importance assessment but also underscores the reliability of our predictive modeling approach.

While the use of LassoNet and SHAP has successfully identified critical features and produced satisfactory regression model fits, there are inherent limitations. Primarily, the feature selection is heavily data-driven, which may not align perfectly with pharmacological interpretations. Additionally, separating the feature selection process from the model training phase could potentially degrade the model's performance.

**Sensitivity analysis.** In this section, we utilize the SHAP model to generate dependence plots and conduct a sensitivity analysis of the four most impactful descriptors, following the approach outlined in [21]. The SHAP model identifies MDEC-23, Lipoaffinity Index, nC (number of carbon atoms), and maxHsOH as the descriptors with the greatest impact on predicting $pIC_{50}$ values. The dependence plot of MDEC-23, Lipoaffinity Index, nC, and maxHsOH descriptors are shown in Figs 8, 9, 10 and 11.

MDEC-23 quantifies the molecular distance between secondary and tertiary carbons, influencing molecular conformation and flexibility, which are crucial for binding to ERα. Higher MDEC-23 values positively correlate with $pIC_{50}$, indicating that increased distance enhances inhibitory activity through optimized receptor engagement. This descriptor has been noted for its role in shaping a molecule's 3D conformation, which can affect not only

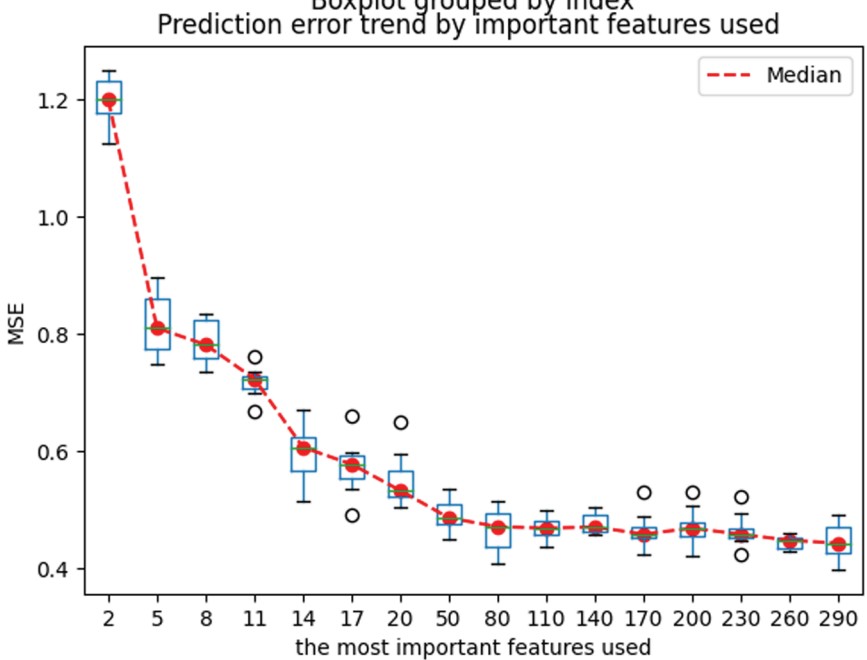

**Fig 5. MSE changes with the number of important descriptors used.**

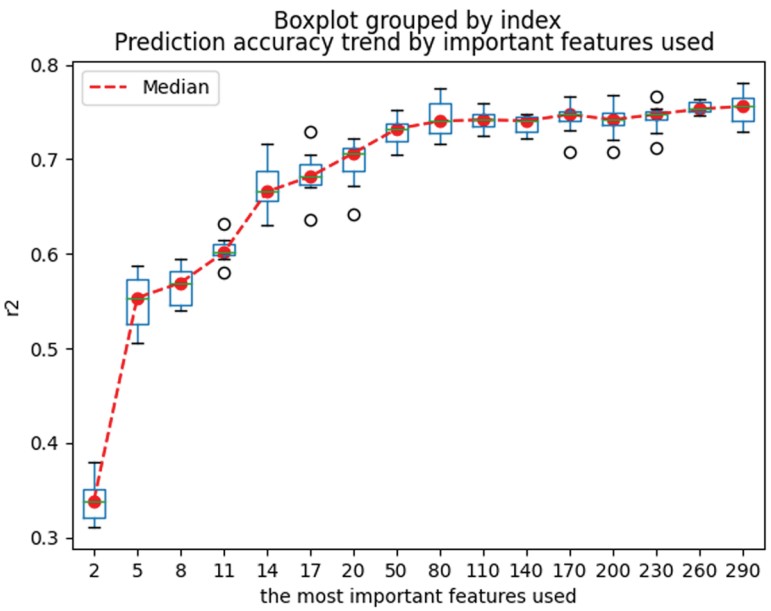

**Fig 6. $R^2$ changes with the number of important descriptors used.**

**Table 2. Standard deviation and mean of $R^2$ for different control groups.**

|  | group 1 | group 2 | group 3 | group 4 | group 5 |
|---|---|---|---|---|---|
| mean | 77.2% | 75.2% | 73.9% | 70.6% | 65.6% |
| standard deviation | 0.040 | 0.037 | 0.042 | 0.036 | 0.053 |

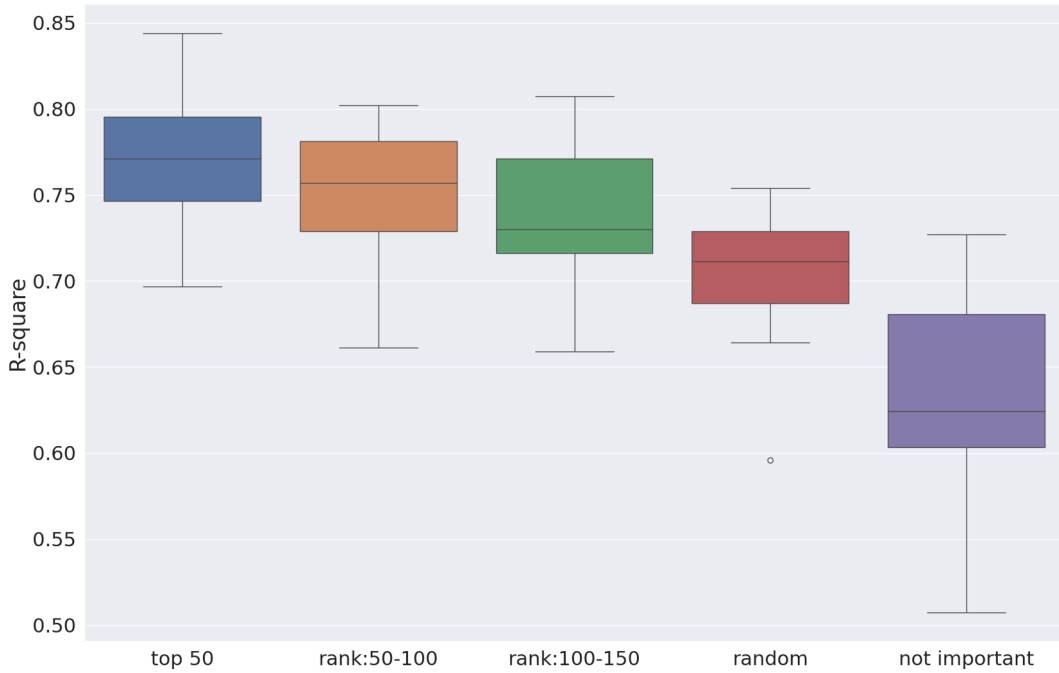

**Fig 7. The performance of different control groups.**

receptor binding but also membrane permeability, aiding in better absorption and distribution [22,23].

The Lipoaffinity Index assesses lipid solubility, essential for cellular membrane penetration. Enhanced solubility, indicated by higher values, improves bioactivity by facilitating better cellular uptake and effective ERα interaction. However, excessive lipophilicity can increase the risk of toxicity and impact ADMET properties negatively by causing longer retention in fatty tissues, which may affect metabolic clearance [24,25]. This highlights the balance needed between optimal lipid solubility and safety [26].

The descriptor nC denotes the number of carbon atoms, which determines molecular size and shape, crucial for ERα binding. An increase in nC correlates positively with $pIC_{50}$, suggesting that additional carbon atoms can enhance bioactivity by providing structural diversity and increasing the potential for hydrophobic interactions at the receptor site. The correlation of carbon count with bioactivity underscores its importance in both pharmacokinetic and pharmacodynamic considerations.

Additionally, maxHsOH, representing the maximum electronic state of hydrogen in hydroxyl groups, significantly impacts the prediction of $pIC_{50}$ values. As illustrated in the SHAP dependence plot, lower maxHsOH values (<0.2) negatively influence $pIC_{50}$ predictions, likely due to insufficient polarity and suboptimal hydrogen bonding capabilities. In the moderate range (0.2 to 0.6), the negative impact decreases but remains, suggesting that intermediate polarity still leads to less favorable binding conformations. At higher values (>0.6), the

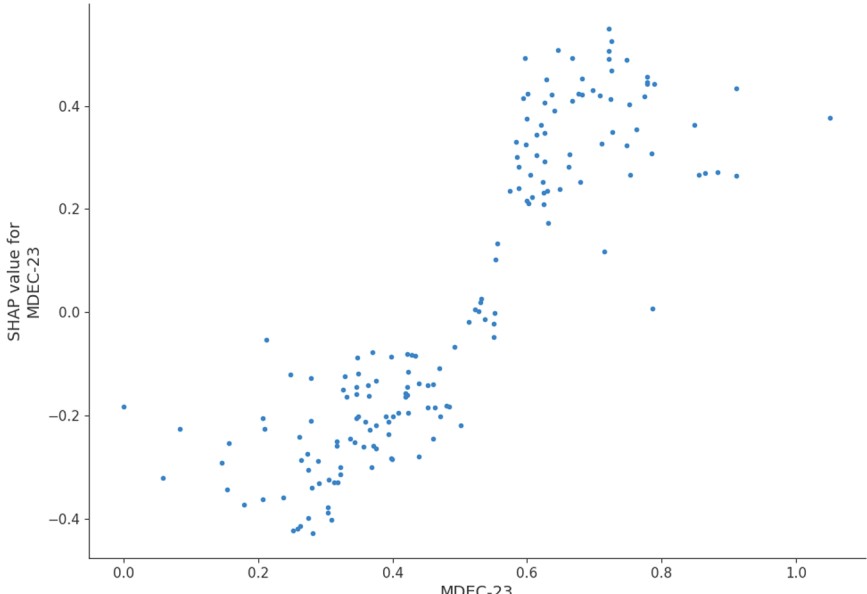

**Fig 8. Dependence plot of MDEC-23.**

influence of maxHsOH stabilizes, indicating that further increases in electronic state do not significantly affect $pIC_{50}$, possibly due to saturation effects or compensatory structural factors. The importance of electronic properties like maxHsOH is highlighted in studies focusing on hydrogen bonding and its role in bioactivity and ADMET behavior [27].

Our further analysis arranges these descriptors in descending order of their influence on bioactivity: maxHsOH, MDEC-23, nC, and Lipoaffinity Index. This ranking underscores the significant roles these descriptors play in predicting compound bioactivity, offering crucial insights for the design of effective ER$\alpha$ inhibitors.

**Comparison of different existing methods.** In this section, we present the comparison results of our proposed workflow against several existing methods.

As shown in Table 3, **Method 3** achieves the worst performance, with an $R^2$ value of less than 0. This indicates that the model performs worse than simply using the average value of all dependent variable values as the prediction. This result highlights the limitations of using all descriptors without feature selection and demonstrates that linear models are not strong enough to capture the complex relationships between molecular descriptors and bioactivity. In **Method 4**, LASSO is used to select the 50 most important descriptors before applying a linear regression model. The performance improves significantly, achieving an $R^2$ value of 53.4%. This suggests that selecting key descriptors is beneficial for improving the predictive ability of regression models. Both **Method 1** and **Method 5** employ neural networks to model the relationship between selected descriptors (or principal components) and bioactivity. They achieve $R^2$ values of 64.6% and 60.2%, respectively. While these results are better than those of linear models, they are still worse than those of tree-based ensemble models. This could be due to several reasons. First, **Method 1** does not optimize or validate the number of important descriptors selected, which could result in suboptimal feature selection. Second, **Method 5**, which uses principal components, may lose critical information during dimensionality reduction, thereby affecting prediction accuracy. Furthermore, neural networks are prone to overfitting and can be challenging to train effectively due to their complex structure. Additionally,

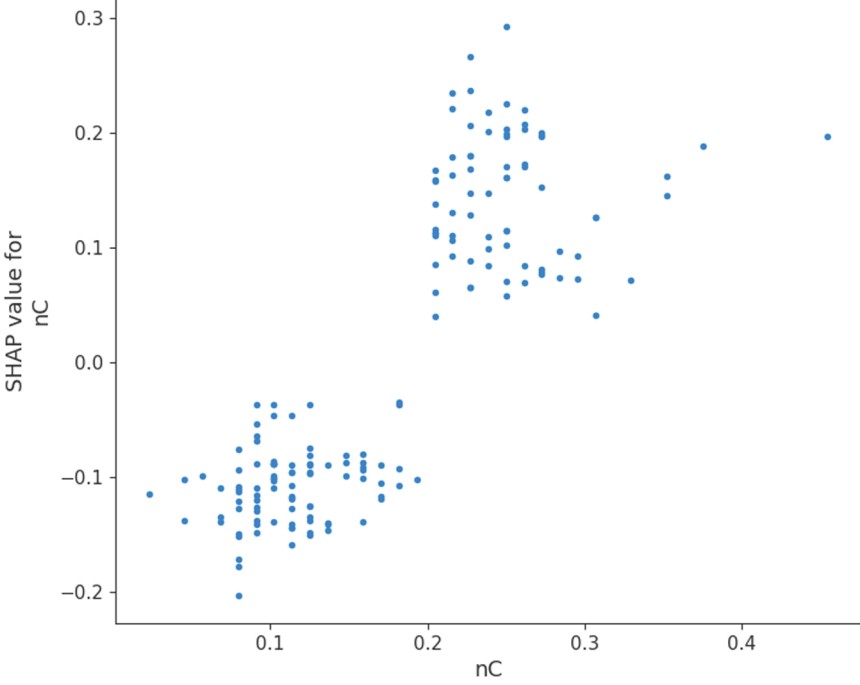

**Fig 9. Dependence plot of nC.**

**Table 3. Performance comparison between different methods.**

|            | Method 1 | Method 2 | Method 3 | Method 4 | Method 5 | Ours   |
|------------|----------|----------|----------|----------|----------|--------|
| $R^2$ Mean | 64.6%    | 73.9%    | -3.10    | 53.4%    | 60.2%    | 77.2 % |
| $R^2$ Std  | 0.0419   | 0.02     | /        | 0.0375   | 0.092    | 0.040  |

*Note: The linear regression model (Method 3) has an analytical solution, and the results are the same each time it is run. Therefore, there is no need to repeat the experiment 20 times to calculate the mean and standard deviation.*

existing literature [20] shows that tree-based methods often outperform deep learning models on tabular data. **Method 2** uses gradient boosting models to fit the relationship between all descriptors and bioactivity, achieving better results than both neural networks and linear models. However, using all descriptors introduces redundancy, which can negatively impact the model's precision and generalization.

Unlike these methods, our proposed workflow fully utilizes variable selection techniques to filter out the most important descriptors and validate their relevance. We then integrate these selected descriptors with a tree-based ensemble model to predict bioactivity, achieving the best performance with an $R^2$ value of 77.2%. This demonstrates the effectiveness of combining feature selection and tree-based models for handling tabular data in bioactivity prediction tasks.

## Prediction of ADMET properties

In this analysis, we compare the performance of the LightGBM single-output model with the XGBoost multi-output model for predicting five ADMET properties, labeled as $Y_1$, $Y_2$, $Y_3$,

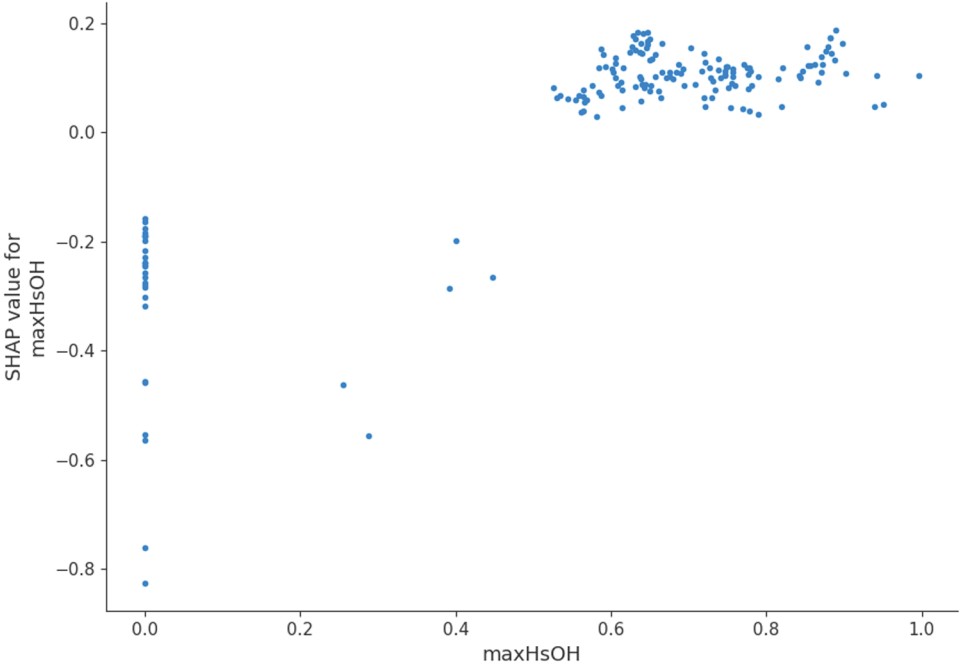

**Fig 10. Dependence plot of maxHsOH.**

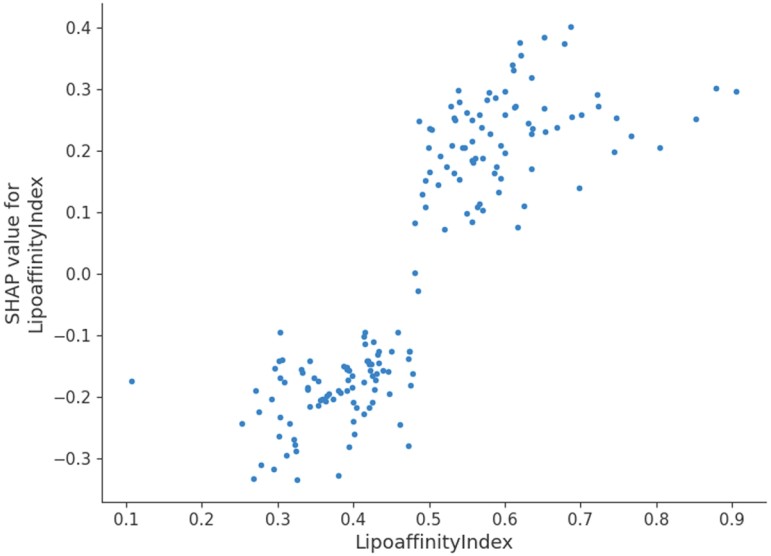

**Fig 11. Dependence plot of Lipoaffinity index.**

$Y_4$, and $Y_5$. For this comparison, these dependent variables are treated as independent. The Mann-Whitney U test is employed to assess the prediction accuracy differences between these two models, with each ADMET property evaluated separately.

The results, presented in Table 4, Table 5, and Fig 12, reveal no significant differences in predictive accuracy between the LightGBM and XGBoost models at the 95 % confidence

**Table 4. The result of Mean-Whitney U test t a 95 % confidence level.**

|  | $Y_1$ | $Y_2$ | $Y_3$ | $Y_4$ | $Y_5$ |
|---|---|---|---|---|---|
| Statistic value | 40.0 | 34.0 | 40.5 | 56.0 | 59.0 |
| p-value | 0.46 | 0.23 | 0.49 | 0.67 | 0.52 |

**Table 5. Comparison of multi-output and single-output classification model across Y1 to Y5**

|  | Y1 | Y2 | Y3 | Y4 | Y5 |
|---|---|---|---|---|---|
| XGBoost ACC Mean | 0.902 | 0.937 | 0.895 | 0.873 | 0.958 |
| XGBoost ACC Std | 0.009 | 0.020 | 0.023 | 0.017 | 0.022 |
| XGBoost ROC-AUC Mean | 0.901 | 0.936 | 0.895 | 0.873 | 0.955 |
| XGBoost ROC-AUC Std | 0.009 | 0.020 | 0.023 | 0.017 | 0.022 |
| LGB ACC Mean | 0.896 | 0.909 | 0.893 | 0.830 | 0.935 |
| LGB ACC Std | 0.012 | 0.031 | 0.030 | 0.036 | 0.028 |
| LGB ROC-AUC Mean | 0.893 | 0.931 | 0.898 | 0.825 | 0.930 |
| LGB ROC-AUC Std | 0.023 | 0.017 | 0.020 | 0.034 | 0.028 |

**Fig 12. Accuracy results comparison of multi-output and single-output classification model.**

level. The statistical values and p-values are shown in Table 4. Furthermore, as detailed in Table 5, the mean ROC-AUC values for $Y_2$, $Y_3$, and $Y_5$ differ by less than 0.03, suggesting negligible differences between the models for these properties. However, for $Y_4$, the models exhibit some variance in performance, as indicated by the distinct ROC-AUC means and standard deviations reported. This suggests that for most properties, either model could be used interchangeably without significant loss of accuracy.

Given the statistical evidence suggesting comparable efficacy, we choose to use the XGBoost multi-output model for practicality and efficiency. This model allows for simultaneous prediction of the five ADMET properties, streamlining the computational process and reducing overall complexity.

## Finding optimal descriptor and best compound

In this study, we employ LightGBM and XGBoost models, configured with the geatpy package, and optimized via a Genetic Algorithm (GA) to maximize the $pIC_{50}$ values for potential anti-breast cancer drugs. The GA is implemented with a substantial initial population size of 500 and is permitted to evolve over 5000 generations. We adopted a high mutation rate of 0.85 and a crossover rate of 0.8 to facilitate effective recombination of genetic material, enhancing both genetic diversity and robustness in the exploration of the solution space. The encoding method used, 'RI' (Real and Integer Mixed Encoding), allows for a flexible representation of both continuous and discrete variables, which is essential for modeling the complex nature of chemical compounds.

The mutation rate was later adjusted to 0.5 to balance the exploration with exploitation, as the algorithm approaches saturation after approximately 2000 generations with limited improvement in the subsequent 3000 generations, as illustrated in Fig. 13. This saturation indicates that the GA has effectively explored potential solutions, optimizing the search for compounds with high bioactivity against ERα. The highest $pIC_{50}$ value identified by the GA is 10.05, which is notably higher than the peak values observed in the dataset (9.860, 10, and 10.337). The robustness of our GA configuration, facilitated by the geatpy toolkit, ensures a thorough optimization process and confirms the effectiveness of the selected genetic parameters.

Given the practical challenges of matching the specific descriptor values found by the optimization algorithm to actual chemical structures, we seek the closest matching compounds from the dataset as a practical solution. To contextualize the results, we assume the optimal solution obtained by the GA as $\hat{x}$ and define the distance between two descriptors $x_1$ and $x_2$ using the Euclidean norm:

$$dist = \|x_1 - x_2\|_2 \tag{6}$$

The three compounds closest to $\hat{x}$ in the dataset, numbered 1867, 1864, and 1865, showcase $pIC_{50}$ values of 9.698, 9.522, and 9.698, respectively. Detailed information on these compounds is provided in Table 6. These results validate the effectiveness of the Genetic Algorithm, demonstrating its capability in identifying highly active compounds.

These findings underscore the utility of integrating machine learning models with optimization algorithms to enhance the drug discovery process. By systematically exploring a broad range of descriptor values and pinpointing the most promising compounds, this methodology offers a robust framework for developing new therapeutic agents targeting ERα in breast cancer. This approach not only streamlines the drug discovery process but also enhances the probability of discovering highly potent compounds.

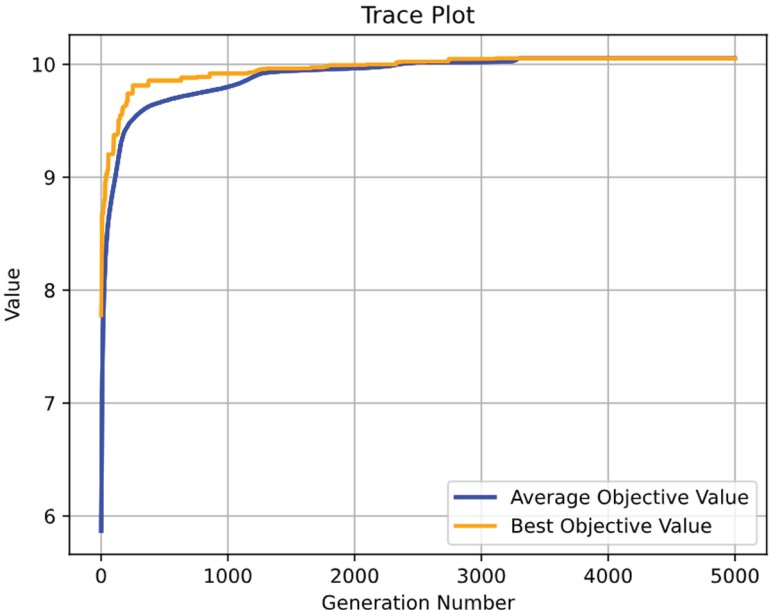

**Fig 13. The trace plot of Genetic Algorithm.**

**Table 6. 3 best candidate compounds in the dataset found by GA.**

| SMILES | No. | $pIC_{50}$ |
|---|---|---|
| Oc1ccc2C(=C(CCCc2c1F)c3ccc(OC(F)F)cc3F)c4ccc(O[C@H]5CCN(CCCF)C5)cc4 | 1865 | 9.698 |
| Cc1oc2cc(ccc2n1)C3=C(c4ccc(O[C@H]5CCN(CCCF)C5)cc4)c6ccc(O)cc6CCC3 | 1864 | 9.522 |
| COc1ccc(C2=C(c3ccc(O[C@H]4CCN(CCCF)C4)cc3)c5ccc(O)cc5CCC2)c(C)c1 | 1867 | 9.698 |

**Table 7. The number of different descriptor types (groups).**

| Descriptor type | Number | Descriptor example |
|---|---|---|
| Acidic group count | 1 | nAcid |
| ALOGP [26] | 3 | ALogP, ALogp2, AMR |
| APol | 1 | apol |
| Aromatic atoms count | 1 | naAromAtom |
| Aromatic bonds count | 1 | nAromBond |
| Atom count | 14 | nAtom, nHeavyAtom, nH, nB, nC, nN, nO, nS, nP, nF, nCl |
| Autocorrelation (charge) | 5 | ATSc1, ATSc2, ATSc3, ATSc4, ATSc5 |
| Autocorrelation (mass) | 5 | ATSm1, ATSm2, ATSm3, ATSm4, ATSm5 |
| Autocorrelation (polarizability) | 5 | ATSp1, ATSp2, ATSp3, ATSp4, ATSp5 |
| Basic group count | 1 | nBase |
| BCUT [28] | 6 | BCUTw-1l, BCUTw-1h, BCUTc-1l, BCUTc-1h, BCUTp-1l |
| Bond count | 10 | nBonds, nBonds2, nBondsS, nBondsS2, nBondsS3, nBondsD |
| BPol | 1 | bpol |
| Carbon types | 9 | C1SP1, C2SP1, C1SP2, C2SP2, C3SP2, C1SP3, C2SP3 |
| Chi chain [29] | 10 | SCH-3, SCH-4, SCH-5, SCH-6, SCH-7, VCH-3 |
| Chi cluster [29] | 8 | SC-3, SC-4, SC-5, SC-6, VC-3, VC-4, VC-5, VC-6 |
| Chi path cluster [29] | 6 | SPC-4, SPC-5, SPC-6, VPC-4, VPC-5, VPC-6 |
| Chi path [29] | 16 | SP-0, SP-1, SP-2, SP-3, SP-4, SP-5, SP-6, SP-7 |
| Crippen logP and MR [24] | 2 | CrippenLogP, CrippenMR |
| Eccentric connectivity index [30] | 1 | ECCEN |

(*Continued*)

**Table 7.** (Continued)

| | | |
|---|---|---|
| Atom type electrotopological state [31] | 488 | nHBd, nwHBd, nHBa, nwHBa, nHBint2 |
| Extended topochemical atom [32] | 43 | ETA_Alpha, ETA_AlphaP, ETA_dAlpha_A |
| FMFDescriptor [33] | 1 | FMF |
| Fragment complexity [34] | 1 | fragC |
| Hbond acceptor count | 4 | nHBAcc, nHBAcc2, nHBAcc3, nHBAcc_Lipinski |
| Hbond donor count | 2 | nHBDon, nHBDon_Lipinski |
| HybridizationRatioDescriptor | 1 | HybRatio |
| Kappa shape indices | 3 | Kier1, Kier2, Kier3 |
| Largest chain | 1 | nAtomLC |
| Largest Pi system | 1 | nAtomP |
| Longest aliphatic chain | 1 | nAtomLAC |
| Mannhold LogP [25] | 1 | MLogP |
| McGowan volume [35] | 1 | McGowan_Volume |
| Molecular distance edge [22] | 19 | MDEC-11, MDEC-12, MDEC-13, MDEC-14, MDEC-22, |
| Molecular linear free energy relation [36] | 6 | MLFER_A, MLFER_BH, MLFER_BO, MLFER_S, MLFER_E |
| Petitjean number | 1 | PetitjeanNumber |
| Ring count | 34 | nRing, n3Ring, n4Ring, n5Ring, n6Ring, |
| Rotatable bonds count | 1 | nRotB |
| Rule of five | 1 | LipinskiFailures |
| Topological polar surface area [23] | 1 | TopoPSA |
| Van der Waals volume | 1 | VABC |
| Vertex adjacency information (magnitude) | 1 | vAdjMat |
| Weight | 1 | MW |
| Weighted path [27] | 5 | WTPT-1, WTPT-2, WTPT-3, WTPT-4, WTPT-5 |
| Wiener numbers [37] | 2 | WPATH, WPOL |
| XLogP [38] | 1 | XLogP |
| Zagreb index | 1 | Zagreb |
| Charged partial surface area [39] | 29 | PPSA-1, PPSA-2, PPSA-3, PNSA-1 |
| Gravitational index [40] | 9 | GRAV-1, GRAV-2, GRAV-3, GRAVH-1, GRAVH-2 |
| Length over breadth | 2 | LOBMAX, LOBMIN |
| Moment of inertia | 7 | MOMI-X, MOMI-Y, MOMI-Z, MOMI-XY, MOMI-XZ |
| Petitjean shape index [41] | 2 | topoShape, geomShape |
| WHIM | 85 | Wlambda1.unity, Wlambda2.unity, Wlambda3.unity |

**Table 8. The 50 selected descriptors for modeling.**

| Number | Descriptor Name | Descriptor Type |
|---|---|---|
| 1 | ALogP | ALOGP |
| 10 | nC | Atom count |
| 11 | nN | Atom count |
| 20 | ATSc1 | Autocorrelation (charge) |
| 22 | ATSc3 | Autocorrelation (charge) |
| 37 | BCUTw-1h | BCUT |
| 38 | BCUTc-1l | BCUT |
| 39 | BCUTc-1h | BCUT |
| 40 | BCUTp-1l | BCUT |
| 41 | BCUTp-1h | BCUT |
| 55 | C1SP2 | Carbon types |
| 57 | C3SP2 | Carbon types |
| 69 | VCH-5 | Chi chain |
| 82 | SPC-6 | Chi path cluster |
| 98 | VP-4 | Chi path |

(*Continued*)

**Table 8.** (Continued)

| 99 | VP-5 | Chi path |
|---|---|---|
| 102 | CrippenLogP | Crippen logP and MR |
| 153 | ndssC | Atom type electrotopological state |
| 290 | SsOH | Atom type electrotopological state |
| 350 | minHBint5 | Atom type electrotopological state |
| 355 | minHBint10 | Atom type electrotopological state |
| 356 | minHsOH | Atom type electrotopological state |
| 386 | mindsCH | Atom type electrotopological state |
| 405 | minsssN | Atom type electrotopological state |
| 409 | minsOH | Atom type electrotopological state |
| 411 | minssO | Atom type electrotopological state |
| 464 | maxHBa | Atom type electrotopological state |
| 465 | maxwHBa | Atom type electrotopological state |
| 474 | maxHBint10 | Atom type electrotopological state |
| 475 | maxHsOH | Atom type electrotopological state |
| 528 | maxsOH | Atom type electrotopological state |
| 530 | maxssO | Atom type electrotopological state |
| 586 | LipoaffinityIndex | Atom type electrotopological state |
| 613 | ETA_BetaP | Extended topochemical atom |
| 621 | ETA_BetaP_ns_d | Extended topochemical atom |
| 638 | nHBAcc | Hbond acceptor count |
| 641 | nHBAcc_Lipinski | Hbond acceptor count |
| 646 | Kier2 | Kappa shape indices |
| 647 | Kier3 | Kappa shape indices |
| 651 | MLogP | Mannhold LogP |
| 658 | MDEC-23 | Molecular distance edge |
| 660 | MDEC-33 | Molecular distance edge |
| 663 | MDEO-11 | Molecular distance edge |
| 664 | MDEO-12 | Molecular distance edge |
| 672 | MLFER_A | Molecular linear free energy relation |
| 701 | nFG12Ring | Ring count |
| 702 | nTRing | Ring count |
| 715 | TopoPSA | Topological polar surface area |
| 722 | WTPT-5 | Weighted path |
| 725 | XLogP | XLogP |

## Conclusion

This study has successfully demonstrated how the integration of explainable AI, machine learning models, and genetic algorithms can enhance the prediction of bioactivity and ADMET properties of compounds targeting estrogen receptor alpha (ERα) in breast cancer therapy. Our methodology not only enhances the accuracy of these predictions but also ensures transparency and interpretability—key aspects that are crucial for advancing both research and clinical applications. Specifically, the use of SHAP and LassoNet for descriptor selection, coupled with genetic algorithms for optimizing descriptor values, has proven effective in identifying compounds with high therapeutic potential. These advances open promising avenues for developing new treatments for breast cancer, underscoring the critical role of sophisticated computational techniques in the drug discovery process. Additionally, our workflow pipeline is not only tailored for enhancing the efficacy of ERα-targeted compounds in breast cancer therapy but can also be generalized to other diseases with similar dataset structures.

Looking forward, it is crucial to validate these predictive models through clinical trials and consider incorporating more diverse datasets. The results derived from data mining must be experimentally verified to ensure that drug development efforts are meaningful. Despite the promising results, this study has several limitations. First, the reliance on a dataset specifically tailored to estrogen receptor-targeting compounds may limit the generalizability of the models to other breast cancer phenotypes, which are characterized by complex and diverse molecular interactions. Additionally, while our computational models are powerful, they depend on the assumption that the dataset descriptors are both comprehensive and capable of accurately representing the relevant chemical space. Currently, our models primarily predict bioactivity based on molecular descriptors, adhering to a 1D-QSAR framework. However, 1D-QSAR has inherent limitations. For instance, when the biochemical mechanisms of molecular properties are unclear, designing effective molecular descriptors becomes challenging, often leading to the failure of QSAR model construction. Since molecular properties are largely determined by their structure—such as the presence and arrangement of functional groups—there is a growing interest in incorporating molecular bonding relationships into QSAR modeling. 2D-QSAR models, which account for these structural relationships, can provide more accurate predictions in certain scenarios. However, excluding potentially significant descriptors not present in the dataset can still lead to biased or incomplete predictions. Moreover, while explainable AI techniques improve model interpretability, they may not fully capture the intricate interactions between molecular descriptors, potentially leading to oversimplified representations of the underlying mechanisms. Finally, the efficacy of genetic algorithms, while beneficial, is highly dependent on specific parameter settings and configurations, which may not be universally optimal across all compound types. Future efforts should focus on addressing these limitations by expanding the dataset, incorporating higher-dimensional QSAR modeling, and fine-tuning algorithmic parameters to improve the robustness and applicability of the predictive models.

## Author contributions

**Conceptualization:** ZeonLung Pun.

**Data curation:** Qiaoyun Xue.

**Formal analysis:** ZeonLung Pun.

**Investigation:** ZeonLung Pun, Qiaoyun Xue.

**Methodology:** ZeonLung Pun.

**Writing – original draft:** ZeonLung Pun, Yichi Zhang.

**Writing – review & editing:** Yichi Zhang.

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
