## [Decision Letter · Decision Letter 0]

29 Dec 2024

PONE-D-24-51400Enhancing ERα-Targeted Compound Efficacy in Breast Cancer Threapy with ExplainableAI and GeneticAlgorithmPLOS ONE

Dear Dr. Pun,

Thank you for submitting your manuscript to PLOS ONE. After careful consideration, we feel that it has merit but does not fully meet PLOS ONE’s publication criteria as it currently stands. Therefore, we invite you to submit a revised version of the manuscript that addresses the points raised during the review process.

Please submit your revised manuscript by   Feb 12 2025 11:59PM. If you will need more time than this to complete your revisions, please reply to this message or contact the journal office at plosone@plos.org. Please include the following items when submitting your revised manuscript:

We look forward to receiving your revised manuscript.

Kind regards,

Manikkam Rajalakshmi

Academic Editor

PLOS ONE

Journal Requirements:

The name of the colleague or the details of the professional service that edited your manuscriptA copy of your manuscript showing your changes by either highlighting them or using track changes (uploaded as a *supporting information* file)A clean copy of the edited manuscript (uploaded as the new *manuscript* file)”

Reviewers' comments:

Reviewer's Responses to Questions

**Comments to the Author**

1. Is the manuscript technically sound, and do the data support the conclusions?

Reviewer #1: Yes

Reviewer #2: Yes

2. Has the statistical analysis been performed appropriately and rigorously? 

Reviewer #1: Yes

Reviewer #2: Yes

3. Have the authors made all data underlying the findings in their manuscript fully available?

Reviewer #1: Yes

Reviewer #2: Yes

4. Is the manuscript presented in an intelligible fashion and written in standard English?

Reviewer #1: Yes

Reviewer #2: Yes

5. Review Comments to the Author

Reviewer #1: Authors proposed "Enhancing ERα-Targeted Compound Efficacy in Breast Cancer Threapy with ExplainableAI and GeneticAlgorithm" the structure of the article is well structured. but authors should follow the following comments.

1.Proofread the entire manuscript.

2. Draw a graphical abstract of your proposed approach

3. compare your approach with existing approaches.

Reviewer #2: The manuscript entitled “Enhancing ERα-Targeted Compound Efficacy in Breast Cancer Threapy with ExplainableAI and GeneticAlgorithm” has many mistakes, authors need to rectify many portions.

• Is the connection between ERα absence and decreased Cyclin D1, PCNA, and TGFβ levels accurately supported by the cited studies?

• Is the term "traditional QSAR methods" appropriately defined, or should it be clarified for readers unfamiliar with QSAR modeling? Is it required IC50 or biological activity initially.

• Please explain AI and Genetic Algorithms in the proposed workflow adequately?

• Are the limitations of the discussed studies, such as over-reliance on specific software or lack of detailed criteria, presented accurately and objectively?

• Is "question D of the 18th Chinese Graduate Mathematical Modeling Contest" the correct source of the dataset, or should it be clarified further?

• Is the website link format correct and functional, as it lacks a clear structure (e.g., "www.shumo.com/wiki/doku.php")?

• Are "5 key ADMET properties" adequately defined for the target audience? Should explanations of binary classifications (e.g., "1" or "0") be elaborated further for clarity?

• What are the methods (SHAP and LassoNet) adequately introduced for a reader unfamiliar with these techniques, or should more context be provided about their use cases?

• Are the machine learning models (e.g., MLP, SVM, LightGBM) described in sufficient detail for understanding their roles and differences in predicting bioactivity?

• Should the evaluation metrics (e.g., R², MSE) for model performance be described in more detail to ensure clarity?

Good Luck!

6. PLOS authors have the option to publish the peer review history of their article (what does this mean?). If published, this will include your full peer review and any attached files.

Reviewer #1: No

Reviewer #2: **Yes: **Shahzaib Ahamad

---

## [Author Response · Author response to Decision Letter 1]

6 Jan 2025

We sincerely thank the reviewers for their valuable time and effort in providing insightful and constructive feedback on our manuscript. Your comments and suggestions have significantly contributed to improving the clarity, rigor, and overall quality of our work. We deeply appreciate your dedication and expertise, which have helped us refine our study and ensure its alignment with the highest academic standards. Once again, we are truly grateful for your thoughtful review and support.

Response to reviewer 1

Q: Proofread the entire manuscript.

A: Thank you for the reviewer’s suggestion. In response, we have carefully proofread the entire manuscript to ensure clarity, consistency, and accuracy. We have also updated the LaTeX template to comply with the PLOS One format. Additionally, we have thoroughly checked our results and ensured the use of unified terminology throughout the manuscript to improve readability and professionalism.

Q: Draw a graphical abstract of your proposed approach

A: Thank you for the reviewer’s suggestion. In response, we have provided a flowchart (Figure 1) that illustrates our proposed workflow, which serves as a graphical abstract. The flowchart outlines the four main components of our workflow: (1) molecular descriptor selection using SHAP and LassoNet, (2) bioactivity prediction with the LightGBM model, (3) ADMET property prediction, and (4) identification of the optimal compound using a genetic algorithm. We believe this visual representation provides a clear and concise overview of our approach.

Q: compare your approach with existing approaches.

A: Thank you for the reviewer’s insightful suggestion. We have compared our proposed workflow with existing approaches, drawing from the following key literature:

1, Wu J, Kong L, Yi M, Chen Q, Cheng Z, Zuo H, et al. Prediction and screening

model for products based on fusion regression and xgboost classification.

Computational Intelligence and Neuroscience. 2022

2, Singh K, Ghosh I, Jayaprakash V, Jayapalan S. Building a ML-based QSAR

model for predicting the bioactivity of therapeutically active drug class with

imidazole scaffold. European Journal of Medicinal Chemistry Reports.

2024;11:100148.

3, Jhanwar B, Sharma V, Singla R, Shrivastava B. QSAR-Hansch analysis and

related approaches in drug design. Pharmacologyonline. 2011;1:306–344.

4, Daoui O, Elkhattabi S, Chtita S, Elkhalabi R, Zgou H, Benjelloun AT. QSAR,

molecular docking and ADMET properties in silico studies of novel 4, 5, 6,

7-tetrahydrobenzo [D]-thiazol-2-Yl derivatives derived from dimedone as potent

anti-tumor agents through inhibition of C-Met receptor tyrosine kinase. Heliyon.

2021;7(7).

We have provided a brief introduction to these existing methods in the "Prediction of Bioactivity" section and presented the comparative results in the "Comparison of Different Existing Methods" section. The results demonstrate that our proposed workflow outperforms the existing methods by effectively utilizing variable selection techniques to identify the most important descriptors. By integrating these techniques with the LightGBM model, our workflow achieves superior performance, attaining an R-square value of 77.2%.

Response to reviewer 2

Q:Is "question D of the 18th Chinese Graduate Mathematical Modeling Contest" the correct source of the dataset, or should it be clarified further? Is the website link format correct and functional, as it lacks a clear structure (e.g., "www.shumo.com/wiki/doku.php")?

A: Thank you for your question regarding the source of our dataset. The dataset used in this study is publicly available and was provided by the China Association for Science and Technology. It was originally made available as part of a data mining competition organized by the Ministry of Education of China. We have corrected the original URL and ensured its accuracy: https://www.shumo.com/wiki/doku.php?id=%E7%AC%AC%E5%8D%81%E5%85%AB%E5%B1%8A_2021_%E5%85%A8%E5%9B%BD%E7%A0%94%E7%A9%B6%E7%94%9F%E6%95%B0%E5%AD%A6%E5%BB%BA%E6%A8%A1%E7%AB%9E%E8%B5%9B_npmcm_%E8%AF%95%E9%A2%98

For the convenience of readers, we have also made all raw data and code available on GitHub, with the corresponding link provided in the Appendix.

Q: Is the connection between ERα absence and decreased Cyclin D1, PCNA, and TGFβ levels accurately supported by the cited studies?

A:Thank you for the reviewer’s question. Upon review, we acknowledge that the statement regarding the connection between ERα absence and decreased Cyclin D1, PCNA, and TGFβ levels is not directly supported by the cited studies in our paper. This was an oversight on our part. Our intention was to highlight the critical role of ERα in the development of breast cancer therapies, but this particular statement was not relevant to our argument. Therefore, we have removed this statement from the manuscript to better align the discussion with the core focus of our research.

Q: Is the term "traditional QSAR methods" appropriately defined, or should it be clarified for readers unfamiliar with QSAR modeling? Is it required IC50 or biological activity initially.

A: Thank you for the reviewer’s suggestion. The term "traditional QSAR methods" refers primarily to the early approaches that used statistical models, particularly linear models, to correlate molecular structure with biological activity for quantitative analysis. These methods were developed to address the limitations of subjective judgments based solely on chemists' experience. To clarify this for readers, we have added additional details to the Introduction section and introduced the classic Hansch model as an example. Additionally, we have provided a brief explanation of IC50 in the Introduction section to ensure clarity and logical flow.

Q: Please explain AI and Genetic Algorithms in the proposed workflow adequately?

A: Thank you for the reviewer’s insightful comment. In response, we have added detailed explanations of AI and Genetic Algorithms, as well as their integration into the proposed workflow, at the beginning of the Methods section. Below is the explanation we have included:

In this section, we propose a comprehensive workflow that integrates machine learning models and Genetic Algorithms to assess the importance of each Molecular Descriptor, predict the bioactivity and ADMET properties of each component, and identify the most promising compounds for developing drugs for breast cancer therapy.

Compared to traditional QSAR linear models, machine learning models excel in their ability to capture complex, nonlinear relationships between Molecular Descriptors and the bioactivity of each component, thereby achieving higher predictive accuracy. Additionally, the use of explainable AI algorithms enables us to interpret the prediction process, providing insights that break the ‘black box’ nature of machine learning models and enhancing their reliability in drug discovery. The Genetic Algorithm plays a crucial role in this workflow, as it allows the optimization process to consider both bioactivity and ADMET properties simultaneously. By combining the predictions of bioactivity and ADMET models, Genetic Algorithms ensures that the optimal potential compounds identified are not only highly bioactive but also exhibit favorable ADMET profiles, making them more suitable for drug development.

We hope this explanation adequately addresses the reviewer's question.

Q: Are the limitations of the discussed studies, such as over-reliance on specific software or lack of detailed criteria, presented accurately and objectively?

A: Thank you for the reviewer’s insightful question. In response, we have expanded the discussion of limitations in the Conclusion section to address this point more thoroughly. Specifically, we added two key aspects:

1, Experimental Verification: We emphasized that the results obtained through data mining using machine learning must be experimentally verified to ensure their relevance and applicability in drug development. Without experimental validation, the utility of these predictive models may be limited.

2, Framework Limitation: We highlighted that this study relies solely on molecular descriptors for prediction, which falls within the framework of 1D-QSAR. While 1D-QSAR is effective to a certain extent, it does not account for molecular bonding relationships, such as structural and spatial interactions, which are critical in understanding molecular properties. This limitation may impact the predictive accuracy and generalization of the models. Future studies could incorporate higher-dimensional QSAR frameworks, such as 2D- or 3D-QSAR, to address this issue and improve the robustness of the models.

We believe these additions enhance the objectivity and accuracy of our discussion on the limitations of the study. Please let us know if further clarification or additional detail is required.

Q: Are "5 key ADMET properties" adequately defined for the target audience? Should explanations of binary classifications (e.g., "1" or "0") be elaborated further for clarity?

A: Thank you for your valuable feedback. We appreciate your suggestion to ensure that the 5 key ADMET properties and their binary classifications are clearly defined for the target audience. In response, we have revised the corresponding Dataset section of our manuscript to provide detailed definitions for each ADMET property and to elaborate further on the binary classification method. Below is a summary of the changes made:

1, We have clarified the biological significance of each ADMET property:

Caco-2 permeability (Y1): Measures the permeability of a compound across small intestinal epithelial cells.

CYP3A4 metabolism (Y2): Assesses the metabolizability of the compound by CYP3A4, a key metabolic enzyme.

hERG channel interaction (Y3): Evaluates the cardiotoxic potential of a compound.

Human Oral Bioavailability (HOB, Y4): Represents the proportion of the drug absorbed into systemic circulation when administered orally.

Mutagenicity (MN, Y5): Assesses genotoxic potential using the micronucleus test.

2, For each property, we explicitly explained the binary classification system (1 or 0) and its interpretation. For example:

A '1' in Caco-2 permeability indicates good permeability, while a '0' indicates poor permeability.

Similarly, a '1' in hERG channel interaction represents cardiotoxic potential, while a '0' indicates no cardiotoxicity.

By elaborating on these properties and their classification, we aim to make the manuscript more accessible and clear for a broader audience, ensuring that the ADMET properties are well-defined and their importance in drug development is evident.

These updates are reflected in the revised manuscript under the Dataset section. We hope this addresses your concern and provides sufficient clarity for the target audience. Thank you again for your thoughtful and constructive feedback.

Q: What are the methods (SHAP and LassoNet) adequately introduced for a reader unfamiliar with these techniques, or should more context be provided about their use cases?

A: Thank you for your thoughtful comments and suggestions. We have revised the Descriptor Selection With explainable AI model section to provide a more detailed introduction to SHAP and LassoNet for readers unfamiliar with these techniques. Specifically:

1, We have elaborated on SHAP's origins in cooperative game theory and its use in assessing individual feature importance.

2, We have expanded the explanation of LassoNet, highlighting its hybrid approach combining neural networks and LASSO regression, as well as its skip layer mechanism for non-linear feature selection.

3, Additionally, we clarified how these methods complement each other, with SHAP focusing on individual descriptor importance and LassoNet addressing group-level interactions, both of which enhance the robustness and interpretability of our models.

We believe this will make the manuscript more accessible and informative for the target audience. Thank you again for your valuable feedback.

Q: Are the machine learning models (e.g., MLP, SVM, LightGBM) described in sufficient detail for understanding their roles and differences in predicting bioactivity?

A: Thank you for the reviewer’s insightful suggestion. In response, we have added a concise introduction to the machine learning models used in this study, including MLP, SVM, XGBoost, RF, and LightGBM, in the Methods section. This addition provides a brief overview of their strengths, limitations, and suitability for handling molecular descriptor data in the context of bioactivity prediction.

Furthermore, we have compared the performance of these models in the Prediction of Bioactivity section, analyzing their respective outcomes in detail. Based on this comparative analysis, we have explained the rationale for selecting the LightGBM (LGB) model as the final predictive tool for bioactivity, emphasizing its superior performance, computational efficiency, and suitability for the dataset.

We believe these revisions provide sufficient detail to help readers understand the roles and differences of the machine learning models in our study. Thank you again for your valuable feedback.

Q: Should the evaluation metrics (e.g., R², MSE) for model performance be described in more detail to ensure clarity?

A: Thank you for the reviewer’s question. In response, we have expanded the Evaluation Metrics subsection under the Results section to provide a more comprehensive interpretation of the metrics used in this study. Specifically, the evaluation metrics are divided into two categories: those used for regression models (R-square, MSE) and those used for classification models (accuracy, ROC-AUC). For each metric, we now include detailed explanations of its formula, significance, and relevance to the evaluation of model performance. These additions aim to improve clarity and ensure that readers can fully understand the purpose and importance of each metric in assessing the predictive performance of the machine learning models.

---

## [Editor Report · Decision Letter 1]

6 Feb 2025

Enhancing ERα-Targeted Compound Efficacy in Breast Cancer Threapy with ExplainableAI and GeneticAlgorithm

PONE-D-24-51400R1

Dear ZeonLung Pun

We’re pleased to inform you that your manuscript has been judged scientifically suitable for publication and will be formally accepted for publication once it meets all outstanding technical requirements.

Kind regards,

Manikkam Rajalakshmi

Academic Editor

PLOS ONE
---

## [Editor Report · Acceptance letter]

PONE-D-24-51400R1

PLOS ONE

Dear Dr. Pun,

I'm pleased to inform you that your manuscript has been deemed suitable for publication in PLOS ONE. Congratulations! Your manuscript is now being handed over to our production team.

Kind regards,

on behalf of

Dr. Manikkam Rajalakshmi

Academic Editor

PLOS ONE